# Diabetes Detection and Communication among Patients Admitted through the Emergency Department of a Public Hospital

**DOI:** 10.3390/ijerph17030980

**Published:** 2020-02-04

**Authors:** Osuagwu Uchechukwu Levi, Frederick Webb, David Simmons

**Affiliations:** 1Diabetes, Obesity and Metabolism Translational Research Unit, School of Medicine, Western Sydney University, Campbelltown, NSW 2560, Australia; da.simmons@westernsydney.edu.au; 2African Vision Research Institute, University of Kwazulu-Natal, Durban 4001, South Africa; 3School of Medicine, Western Sydney University, Campbelltown, NSW 2560, Australia; 19429364@student.westernsydney.edu.au; 4Campbelltown Hospital Diabetes Services and Western Sydney University, Campbelltown, NSW 2560, Australia

**Keywords:** type 2 diabetes, opportunistic testing, blood glucose, hyperglycemia, emergency department, South Western Sydney Local Health District, public health, hospital admission, Hemoglobin A1c, Discharge summary

## Abstract

Early identification/diagnosis of diabetes and frequent monitoring of hyperglycemia reduces hospitalizations and diabetes-related complications. The present study investigated the proportion of older adults coded with diabetes or newly diagnosed during their admissions and assessed discharge summary content for diabetes-related information. The study used electronic data on 4796 individuals aged ≥60 years admitted through the emergency department (ED) of a public hospital from 2017 to 2018 extracted using International Classification of Diseases, Tenth Revision, Clinical Modification (ICD-10-CM code). The proportion of admitted patients who were diagnosed with diabetes over a one-year period, proportion with glycated hemoglobin A1c (HbA1c) and random blood glucose (RBG) test performed during their stay, length of stay, discharge summary information and the factors associated with elevated HbA1c (>7%/53 mmol/mol) were investigated. In total, 8.6% of ED presentations to the hospital were coded with diabetes, excluding gestational consisting of 879 patients (449 males, 430 females) aged ≥ 60 years (74.6 ± 8.9 years). In total, 98% had type 2 diabetes (n = 863), 53% were Australian-born (n = 467), and the mean body mass index (BMI, 31 ± 7 kg/m^2^; n = 499, 56.8%), RBG (9.8 ± 5.2 mmol/L; n = 824, 93.7%) and HbA1c (8.0 ± 2.0%; n = 137, 15.6%) and length of stay (6.7 ± 25.4 days) were similar between gender, age, and nationality (*p* > 0.05). Three coded patients (0.3%) were newly diagnosed during the admission. In total, 86% had elevated HbA1c, but this was recorded in 20% of discharge summaries. Patients who are on a combination therapy (adjusted odds ratio 23%, 95% confidence intervals: 7%/38%), those on SGLT2 Inhibitors (aOR, 14%: 2%/26%) or had a change in medication (aOR, 40%: 22%/59%) had lower odds of having elevated HbA1c during admission. The low diagnosis rate of diabetes and the lack of clinical assessment of HbA1c in older adults admitted through the ED of a South Western Sydney public hospital suggest that many patients with diabetes either remain undiagnosed even during admission and/or are going to the ED with unknown diabetes that is unidentified with current practices. The clinically important HbA1c results were only infrequently communicated with general practitioners (GPs).

## 1. Introduction

Diabetes is a significant health problem globally [1], in Australia [2] and in the South Western Sydney (SWS) region [3], resulting in significant morbidity and mortality [4]. In 2017, there were approximately 1.3 million adults living in Australia with known diabetes, with the number reaching well over 1.7 million including those with undiagnosed diabetes [2]. In SWS, the number of people with diabetes (all types) has increased by over 158% since 2000 [3] exceeding that of the state and national averages [3,5]. The rising obesity, the ageing population, dietary changes, and sedentary lifestyles [6,7] likely drive this increase. According to a recent monograph, approximately 60.9% of people with diabetes in SWS were aged 60 years and over [8]. Evidence indicates that diabetes in older adults is linked to higher mortality, reduced functional status, and increased risk of hospitalization [9]. Older adults with diabetes are also at substantial risk for both acute and chronic microvascular and cardiovascular complications of the disease [10]. However, under detection remains a major barrier to prevention of diabetes and associated complications particularly in older individuals, who are considered a high-risk group for diabetes [8,11], with more than a third of people with diabetes being undiagnosed on presentation to hospital [12].

Early detection of diabetes is recommended because the duration of hyperglycemia is a predictor of adverse outcomes and there are effective interventions to reduce the risk of complications [13,14]. Although, the recommended laboratory techniques and protocols for diabetes testing are fasting plasma glucose (FPG) level or a two-hour oral glucose tolerance test (OGTT), these are not suitable for acutely ill patients in an emergency department (ED) setting [13]. The World Health Organization (WHO) and the American Diabetes Association recommends that diagnosis and monitoring of people with diabetes can be made with glycated hemoglobin A1c (HbA1c) alone (≥6.5% or 48 mmol/mol on two separate occasions [13,15]), but the test is influenced by the lifespan of the red blood cell, generating multiple false positive and false negative results compared with the OGTT. Thus, it was suggested that HbA1c in combination with blood glucose testing should be used in diabetes screening [16].

There are many individuals with risk factors for diabetes, who do not have access to primary care and present to the ED seeking care for unrelated illnesses. The ED also provides medical care to people with little or no access to screening and preventive interventions; therefore, initiating a screening intervention in the ED may improve the health of this vulnerable population. The ED of Campbelltown hospital of SWS serves approximately 283,743 residents across the district (ABS Census 2016) [17] and during a visit to the ED, blood glucose levels are also regularly tested particularly in high-risk individuals, making the department a potentially valuable location to conduct opportunistic screening for diabetes [12,18,19,20]. Given the large number of presentations at the ED, missed opportunities for diabetes screening could be common [21] and patients who are newly diagnosed with diabetes from the screening test may not have been closely monitored [22]. The aim of this study was to determine the proportion of patients aged ≥ 60 years admitted through the ED coded with diabetes or newly diagnosed during their admission over a one-year period. Furthermore, the study sought to determine the sort of notes and follow-up requests made in the patients’ discharge referrals regarding the management of diabetes.

## 2. Materials and Methods 

### 2.1. Study Setting and Population 

This retrospective cohort study reviewed the clinical data of patients admitted through the ED of a public Hospital in SWS. A list of all admitted patients that were coded with diabetes (excluding gestational diabetes mellitus), between 1 July 2017 and 30 June 2018, was provided by the clinical information department (CID) at the hospital. The electronic file contained some of the required demographic data (age, sex, country of birth) for each patient along with admission details (e.g., dates of presentation and discharge, reason for presentation, length of stay [date of discharge minus date of admission] see Table 1), coded diabetes complications (e.g., diabetic nephropathy or background retinopathy), and co-morbidities (hypertension and ischemic heart disease). The remaining parameters including glycated hemoglobin (HbA1c), discharge summary notes and requests for follow-up were extracted manually from each selected patient’s electronic medical record (Cerner Millennium, Cerner Corporation. North Sydney, NSW, Australia). Where available, glucose results from laboratory and point of care blood glucose testing obtained from the ED using a Roche Cobas 6000 chemistry analyzer (Roche Diagnostics, Indianapolis, IN, USA) and an Abbot I-Stat device with CHEM 8+ cartridges (Abbott point of Care Inc., Princeton, NJ, USA), respectively, were also extracted from patient record. Obesity as coded in the ICD-10-AM was provided by the department and was cross-checked manually in the electronic records of individual patients. Where BMI was available in the patient records, this was preferred, and the classification of obesity was BMI ≥ 30 kg/m^2^. 

### 2.2. Ethics 

The study was considered as a quality assurance activity by the hospital Quality and Safety Office, under the framework overseen by the South Western Sydney Local Health District Human Research Ethics Committee (#QA18/021).

### 2.3. Selection Protocol

The extraction included 5100 electronic admission records of patients who visited the ED within the study period and were coded with diabetes. Duplicate admissions were identified and removed, and the remaining admissions were randomized using an Excel spreadsheet function and manual selection. It was not feasible to go through all the admitted patient electronic medical records, and so randomization was used to select a representative sample from a pool of patient records (all ages). Only the medical records of patients aged ≥ 60 years were reviewed and included in this study. This age group was selected because, in Australia, prevalence in this age group is substantially higher than the average for all age groups [23]. In order to avoid data clustering, only one admission was kept for patients who were admitted more than once. In general, the most recent admission was kept. However, if a patient had an admission with information related to diabetes in their clinical notes or follow-up requests in their discharge referral, or if the admission included a HbA1c measurement, this was kept in preference to any other admission. No new cases of diabetes were excluded due to this selection process.

### 2.4. ED Outcome Variables

Using an ethics approved data collection form, one researcher (FEW) viewed and systematically retrieved required information from the medical record of the selected admissions. To ensure reliability of the data set, a second reviewer (ULO) reviewed the electronic medical records of the first 35 patients and the data extracted was the same as that extracted by the first reviewer. The variables that were extracted included demographic characteristics, presentation details, reported weight, BMI, medical history, vital signs, medications, reason for presentation, time of presentation, mode of transport to ED, transfer to other facilities on discharge, random serum glucose, glycosylated haemoglobin (HbA1c), blood pressure levels, diabetes type, diabetes comorbidities including presence and absence of hypertension, and complications including death. Notes in the discharge summary including request for follow up, referrals, and documentation of blood glucose levels were also examined. HbA1c of 7% or lower or laboratory blood glucose ≤ 12.0 mmol/L was considered target for glycaemic control in the admitted patients in this hospital. 

### 2.5. Statistical Analysis

Notes and requests for follow-ups were categorized and all data was entered into a secure Excel spreadsheet (Microsoft Inc. Redmond, WA, USA) accessible to only the researchers. All statistical analysis was conducted using IBM SPSS Statistics version 25 (IBM Inc. Chicago, IL, USA). The level of statistical significance was set at 5%. Normality distribution assessment confirmed that all assumptions met the parametric text. A chi-square test for discrete variables and one-way analysis of variance for continuous variables were utilized to assess the differences between variables or groups. General linear model for multivariable analysis determined the association of age, gender and ethnicity with other variables and determined the factors associated with elevated HbA1c. Results were presented descriptively as mean (standard deviation), counts and/or proportions and adjusted odds ratio (aOR) for associations between variables, where necessary. 

## 3. Results

Figure 1 presents the flow diagram of the patients’ records and method of selection. Of the 55,968 ED presentations to the hospital during the one-year study period, there were 4796 unique people admitted and coded with diabetes excluding gestational, resulting in a prevalence of 8.6% among patients aged 65 ± 17 years (mean ± SD). The final selection resulted in the inclusion of 879 (18.3%) hospital admissions of people (449 males, 51.1%, and 430, 48.9% females) aged ≥ 60 years (mean age of 74.6 ± 8.9 years), in this study.

### 3.1. Characteristics of the Patients

Table 1 presents the demographic characteristics of the patients in this study including their medication types. The majority of the admitted patients in this study hospital were Australian-born (n = 467, 53.1%, including 23 (2.6%) Aboriginal patients) and European-born (n = 408, 46.4%), who were also significantly older (77 ± 8 years vs 74 ± 9 years, *p* < 0.0001) than the Australian-born patients. Figure 2 presents the breakdown of the participant’s country of birth in this study. 

There were three newly diagnosed patients (0.3%) with diabetes during the study period and their ages were 62, 65, and 85 years. BMI was recorded in 56.8% of the patients (*n* = 499, mean BMI was 31 ± 7 kg/m^2^) and 78.4% were either overweight or obese. There were significantly more people within the 70–79 years (84.6%, *n* = 143/169) and 60–69 years (83.8%, n = 145/173) age groups who were either overweight or obese compared to those aged >80years (65.6%, *n* = 103/157; *p* = 0.0005). 

The mean length of stay during admission was 6.7 ± 25.4 days, with 63% staying longer than one day: this was not dependent on gender, age, ethnicity, HbA1c or RBG levels of the patients. Among this age group, the reasons for presentation to the ED were primarily due to chest pain/abdominal pain (17.6%) and shortness of breath (10.4%) (Figure 3), but for the newly diagnosed patients, hyperglycemia was the main presenting symptom in two of the three patients.

Regarding diabetes medications, 78.6% (n = 691) of the patients with diabetes were taking at least one medication during the period of this study, and 35.3% (n = 310) were taking multiple (>2) medications. Approximately one-half of the patients (n = 444) had at least one diabetes-related complication and similar proportion had at least one diabetes-related comorbidity (49.5%, n = 435) during admission. Figure 4 shows the breakdown of the complications and comorbidities experienced by the patients during admission and the rate of assessment of glucose level using HbA1c test in these patients.

### 3.2. Glycaemic Monitoring of People with Diabetes 

One hundred and thirty-seven patients (15.6%) had at least one HbA1c test during admission, including the 3 newly diagnosed individuals, and this was similar between T1DM and T2DM (21.0% vs. 15.4%, *p* = 0.45) patients. Compared to this, significantly more patients had RBG (n = 857, 97.5%, *p* < 0.0001) which was also similar between T1DM and T2DM 100% vs. 97.5%, *p* = 0.75). The mean values of laboratory and finger prick blood glucose testing were similar (9.8 ± 5.2 mmol/L, versus 10.1 ± 4.7 mmol/L, *p* = 0.146) among admitted patients with diabetes. The proportion of patients aged 60 years and over, who had HbA1c < 7% during admission were significantly fewer than those who had target RBG (≤11.0 mmol/L) (5.9% vs 71.8% for laboratory or 54.0% for finger prick glucose, *p* < 0.05). Among the patients with a glucose level of ≤11.0 mmol/L, who also had an HbA1c test completed during admission (n = 75), forty-three percent (n = 49) had a HbA1c above 7%, indicating the differences between both tests for identifying patients with abnormal glycaemia.

Age and country of birth were associated with the patients’ mode of transportation to ED and of complications, diabetes-related foot ulcer was predominant among the admitted patients who were aged 60–69 years (4.3%) compared to those who were aged 70–79 years (1.3%) and those aged ≥ 80 years and over (0.4%, *p* = 0.002). On discharge from the ED, 17.1% (n = 150) of the patients in this study were transferred to the critical care unit and the rest went to the wards (729, 82.9%). 

### 3.3. Discharge Summaries

For eight hundred and thirty-seven patients (95.2%), there were notes on their discharge summaries regarding diabetes and for the remaining 42 patients (4.8%), there was no note. Table 2 presents the analysis of the notes in the discharge summaries. One in five discharge summaries contained some notes on the participant’s diabetes, including changes to diabetes medication and abnormal blood glucose levels. For the patients with elevated HbA1c, 20% had a note regarding the test result in their discharge summaries, compared to 5.8% (*p* = 0.015) of the patients with HbA1c ≤ 7% (n = 3) with such notes in their discharge summaries.

Further analysis on the type of patients that make up the top note requests (Table 2) showed that younger patients (4.2% for 60-69 years) were more likely to have this noted in their DS, compared to older patients (2.4% and 0.8% for 70–79 years, and ≥80 years old, respectively; *p* = 0.03). Those with elevated HbA1c during admission (20.5%), were also more likely to have this noted in their DS than those with HbA1c ≤ 7% (6.0%, *p* = 0.003). Having a diabetes-related note in the DS was more likely to occur in patients on multiple diabetes medications (33.7%) compared to those on either two (17.7%) or a single medication (16.8%, *p* < 0.0005) for diabetes management. Patients on SGLT2 inhibitors (41.9% vs. 18.7%, *p* < 0.0005), those on DPP4 inhibitors (30.6 vs. 17.7%, *p* < 0.0005), sulfonylureas (32.1 vs. 16.9%, *p* = 0.0001) and insulin (44.7% vs. 11.0%, *p* < 0.0005), were more likely to have diabetes-related information in their DS, compared to those who were not taking any of the medications (*p* < 0.0005, for all comparison) during admission. All patients (100%) with a change in diabetes medication had a diabetes-related note in their DS compared to 8.5% of those that had no change in medication during admission (*p* < 0.0005). Those who were referred to a diabetes educator (100% vs 18.4%, *p* < 0.0005) or to a specialist (27.1% vs. 11.9%, *p* < 0.0005) and those with poor glycaemic control (71.1% vs. 36.0%, *p* < 0.0005), during admission were also more likely to have diabetes-related notes in the DS. 

### 3.4. Factors Associated with Elevated HbA1c 

The odds of having elevated HbA1c during admission was reduced by 14% (95% CI of aOR 2%–26%) and by 23% (95%CI of aOR: 7%–38%) in patients who were taking SGLT2 Inhibitors or a combination therapy consisting of insulin and metformin, respectively, during admission. A change in diabetes medication of the patients was also associated with 40% (95% CI of adjusted odds ratio 22%–59%) reduction in the odds of having elevated HbA1c, during admission. There was no significant association between the demographic variables and having elevated HbA1c in this study. 

## 4. Discussion

The main findings of this study were the low diagnosis rate of diabetes (0.3%) in the admitted patients coded with diabetes, over a one-year period, the low communication rate between the hospitals and the general providers, and the very low uptake of glucose monitoring using HbA1c test, even in those with diabetes-related complications/comorbidities, despite the high prevalence of diabetes in this age group [8] and in the SWS region [25]. A higher prevalence of diabetes (38.4% vs. 8.6%) was found in another public hospital ED in Western Sydney, a neighboring district with a similar high diabetes prevalence, and 32.2% of those with diabetes were previously undiagnosed diabetes of diabetes (32.2%), after 6 months of testing for diabetes in adults using HbA1c, irrespective of their presenting signs [12]. The proportion of newly diagnosed individuals with diabetes increased by 7-fold (from 0.3% to 2.2%); considering only the individuals with HbA1c test done during the admission (n = 3/137), however, this was still lower than in the previous study [12] and the 15% population prevalence of diabetes in those aged ≥ 60 years [8]. Since this region and Western Sydney are the hotspots for diabetes in the country [25], a larger than average number of undiagnosed individuals with diabetes were expected to be living in these areas. The percentage of individuals with diabetes in this study was significantly lower than the expected proportion in the general population, which indicates that there must be people either going to the ED with unknown diabetes that is unidentified with current practices, and/or that a significant proportion of the individuals with diabetes remain undiagnosed throughout their admission. This may be due to the low uptake of HbA1c testing during admission and thus, are prone to having more complications due to the hyperglycaemic duration [13,26]. It appears that this SWS public hospital may not be delivering this much-needed service, even in such high-risk age group [8]. This is despite the various government initiatives encouraging routine screening and monitoring of diabetes in the Australian public hospitals, particularly at the ED, using HbA1c test [27,28], and the evidence favouring screening for diabetes using HbA1c over other screening techniques including Oral Glucose Tolerance testing (OGTT) [27]. The potential medico-legal liability of diagnosing without ensuring follow-up occurs, crowding and staff constraints of the ED and wards, and reluctance of doctors to treat hyperglycemia for fear of causing hypoglycemia are possible barriers to effectively diagnose and prevent diabetes-related complications [29,30]. Initiating an automatic request procedure for HbA1c testing particularly in high-risk groups (older adults and Aboriginals) admitted through the ED is needed to reduce the burden of diabetes across the district.

The fact that the important HbA1c test result or request related to the patient’s diabetes was made only in one of five admitted patient records indicate the need to measure and improve discharge summary quality in this hospital. Information in the discharge summary helps the general practitioner in decision making including altering a patient’s management plan, and/or referring to specialist (endocrinologist, dietician or diabetes educator). In this study, patients aged between 60 and 69 years referred to a diabetes specialist for inadequate glycaemic control were more likely to have these recorded in their discharge summary notes. For example, in all three patients diagnosed with diabetes during their admission, this was in their discharge summaries, along with newly prescribed medications. Ideally, a request for follow-up should be found in the discharge referral of patients diagnosed with diabetes or those found to have blood glucose or HbA1c results suggestive of diabetes, and absence of such records may compromise the ability for treatment and prevention of complications [18,29,30], which may result in a higher risk of re-admission [31]. 

Although, HbA1c has been used widely to assess glycaemic control in patients with diabetes [32], the test was rarely used in this hospital in people with both T1DM and T2DM. Even during the admission, not many patients with diabetes seem to have completed HbA1c test and no notes in the patients charts to indicate why the test was not completed. Among the patients who had HbA1c test done, the readings were elevated in 77% of the cases but the odds were reduced in those who were taking a combination therapy (insulin plus metformin), or the SGLT2 inhibitors. Studies have shown that HbA1c results correlate with diabetic microvascular disease such that reduction in HbA1c reduces the risk of complications in people with diabetes [33]. Also, the fact that only a few people were identified as having T1DM in this age group suggests the low specificity value of ICD codes for T1DM (26%) in previous study [34].

Overall, the patients in this study spent more nights in the hospital than was previously reported among inpatient with (7 vs. 5–6 days) and without (approximately 5 days) diabetes [35,36], which increases the cost of hospitalization [37,38]. A study found that the cost of hospitalization was approximately 1.5 times higher in patients who were aged ≥ 60 years than those aged ≤ 40 years. The cost further increases if the patients had any diabetes-related complications [39] and in the present study, a large proportion of the admitted patients (Figure 4) had a complication which further indicates the high burden of diabetes in this public hospital. Formal early detection of diabetes through opportunistic testing in this high-risk age group population will help to reduce the burden of diabetes by identifying new cases and reduce complications. Past studies [12,25,40] including a recent study in Blacktown Hospital which identified 32.2% new adult cases of diabetes following a six-week screening period using HbA1c testing [12] has shown significant benefits of opportunistic testing in hospitals. In another study, George et al. found 2.6% new cases of diabetes through opportunistic testing of 500 ED patients without previous diabetes [41] which was similar to the 2.2% found in this study considering only those who underwent HbA1c testing.

There are some limitations to this study. The study relied on accurate clinical coding for new diabetes diagnosis, which could have underestimated the number of newly diagnosed diabetes at the hospital by up to 28%, as was suggested in a previous study [12]. Data for some patients with diabetes, who attended the ED and discharged on the same day, may not be captured in this study. The 3 month chart review should have identified all individuals with known diabetes, but in the absence of testing, the study design restricted the data to participants coded with diabetes in the ED, making it difficult to find potentially missed cases with diabetes. In addition, other variables may have been inaccurately coded, and this may affect the reported prevalence of certain diabetes-related complications and co-morbidities. For example, weight record was inconsistent, and found in different locations throughout the medical record and frequently omitted in short admissions or those of older patients. In some cases, the only available measurement was an estimated weight, which when compared with actual measurements in the patients’ files (where both measurements were available), often varied greatly. The finger prick blood glucose levels were often, but not always taken at triage, and the initial reading may not necessarily reflect the patient’s initial blood glucose level on entry to the hospital. In a number of case histories, a patient may have been described as critically hypoglycaemic on arrival to the ED but may have been treated by the time a blood glucose test was done, which would bring the glucose reading within the normal range. The same was true for some patients who arrived at the ED with extreme hyperglycemia or blood pressure outside the normal range.

## 5. Conclusions

In summary, there was a low diagnosis rate of diabetes among older adults admitted to the hospital through the ED in South Western Sydney. The low uptake of HbA1c testing among these individuals indicate that many individuals aged ≥ 60 years with diabetes remain undiagnosed even during admission and/or that many are going to the ED with unknown diabetes that is unidentified with current practices. The clinically important HbA1c result was seldomly communicated to the GPs, suggesting the need to measure and improve discharge summary quality in this hospital. Unavailability of such notes in the discharge summary may affect the management plan of patients or lead to unnecessary test repeats during consultation.

## Figures and Tables

**Figure 1 ijerph-17-00980-f001:**
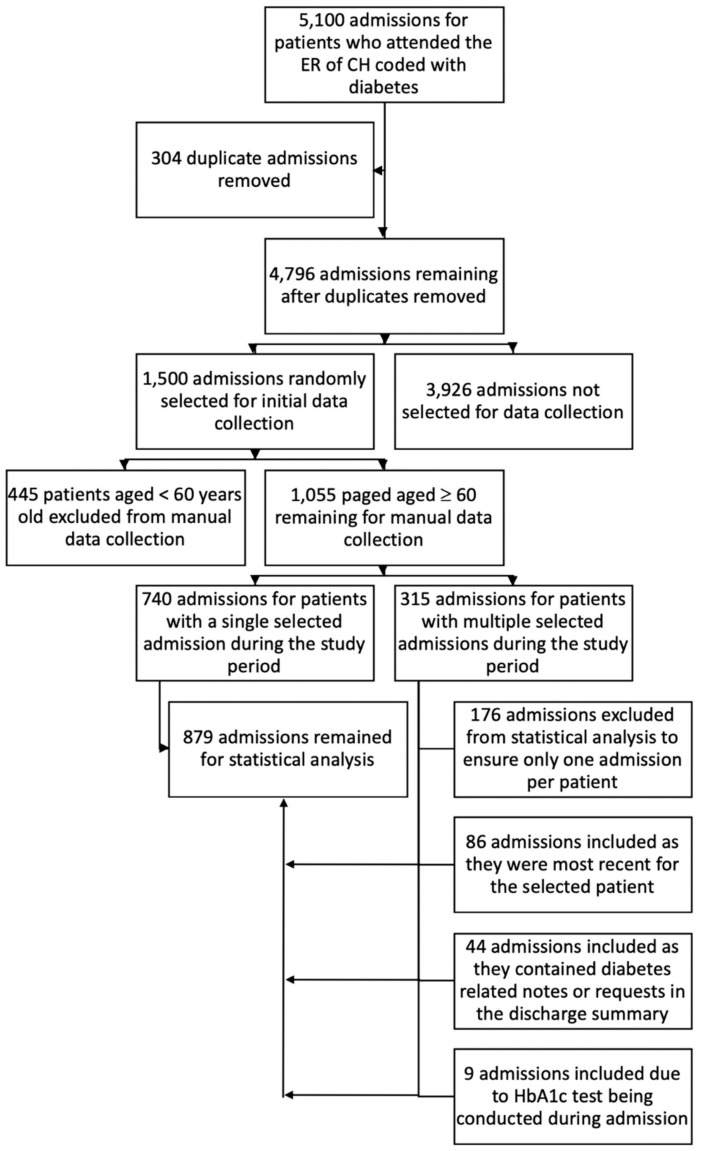
Flow chart for method for admission selected for inclusion in the study.

**Figure 2 ijerph-17-00980-f002:**
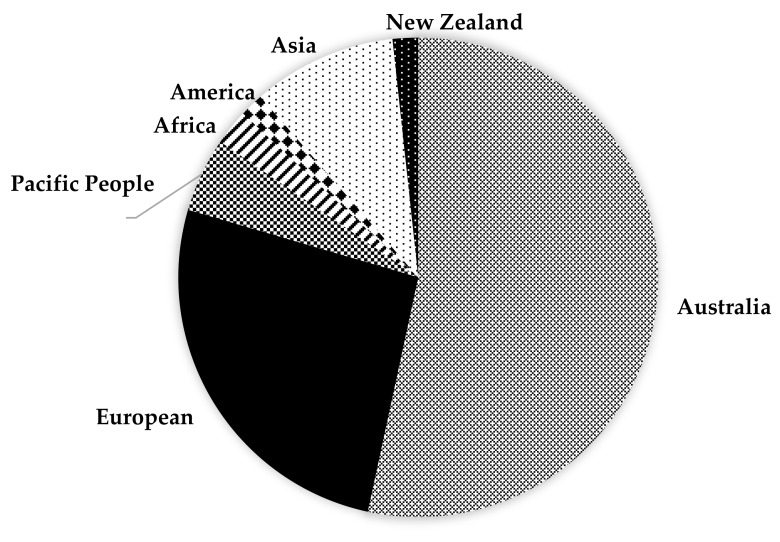
Distribution by country of birth of patients with diabetes admitted through the emergency department.

**Figure 3 ijerph-17-00980-f003:**
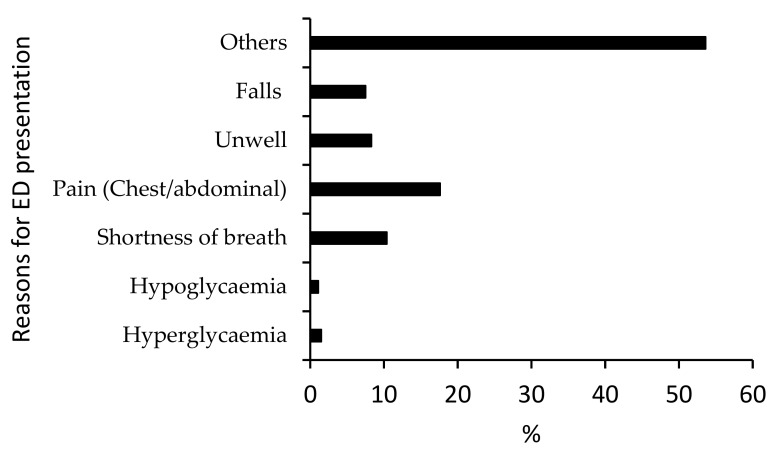
Reasons for presentation to the emergency department among admitted patients with diabetes.

**Figure 4 ijerph-17-00980-f004:**
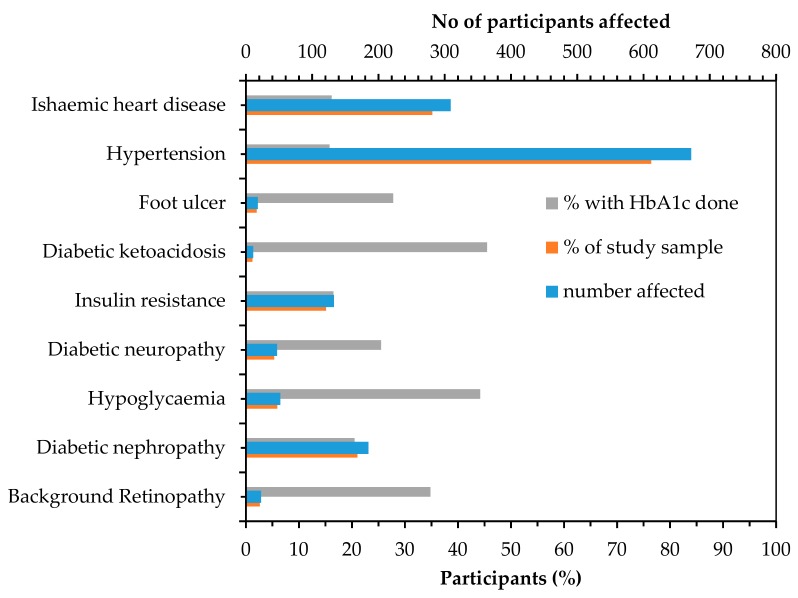
Number of patients, admission rates and rate of glycated hemoglobin A1c (HbA1c) assessment, within patients with diabetes-related complications and comorbidities.

**Table 1 ijerph-17-00980-t001:** Characteristics of patients with diabetes attending the emergency department (ED) between 2017 and 2018 (number n = 879, except where indicated).

Parameter	Frequency/Values
Age at admission, (mean ± Standard deviation SD) years	74.6 ± 8.9
Gender, (n, %)	
Male	449 (51.1)
Country of birth, n (%)	
Australia	467 (53.1)
Overseas	408 (46.4)
Diabetes type, n (%)	
Type 1 diabetes	12 (1.4)
Type 2 diabetes	863 (98.2)
Other types	4 (0.4)
Body mass index (mean ± SD), kg/m^2^	30.5 ± 7.0
Obesity (≥30 kg/m^2^)	238 (47.7)
Weight (mean ± SD), kg (n = 522)	83.8 (20.1)
Blood glucose testing	
HbA1c done, n (%)	137 (15.6)
Mean ± SD %, mmol/mol	8.0 ± 1.9, 63.5 ± 20.8
HbA1c > 7%, n (%) ^α^	85 (62.0)
Random Blood Glucose (RBG) done, n (%) (Laboratory RBG/finger prick RBG)	824 (93.7)/633(70.0)
Mean ±SD lab RBG/finger prick RBG, in mmol/L	9.8 ± 5.2/10.1 ± 4.7
Laboratory, finger prick RBG > 11 mmol/L, n (%)	193 (23.4), 158 (25.0)
Blood pressure (mean ± SD), mmHg	
Systolic	140 (26)
Diastolic	76 (13)
BP >130/80, n (%)	576 (65.5)
Length of stay, (mean ± SD) days	6.7 ± 25.4, range 0–370
>1 day	556 (63.3%)
Arrival by emergency medical service (ambulance)	558 (63.5%)
From nursing home or long-term care facility, %	94 (10.7)
Transfer to critical care unit	150 (17.1%)
Diabetes management, n (%) ^ƛ^	
Monotherapy (one treatment)	300 (34.1)
Combination therapy (>one treatment)	391 (44.5)
No medication	188 (21.4)
Medication type	
Insulin alone	241 (27.4)
Metformin alone	489 (55.6)
Insulin and metformin	103 (11.7)
Other oral hypoglycaemic agents	349 (39.7)
Sulfonylureas	201 (22.9)
DPP4 inhibitors	182 (20.7)
SGLT2 inhibitors	64 (7.3)
Source of referral to ED	
Self/family/friends	608 (69.2)
Medical practitioner (General Practitioner GP/Dentist)	127 (14.4)
Other facilities	144 (16.4)
Mode of transport to ED	
Ambulance	564 (64.2)
Private car/others	261 (29.7)
No transport	54 (6.1)

Abbreviations: HbA1c, glycated hemoglobin; DPP4, dipeptidyl peptidase-4; SGLT2, sodium glucose co-transporter protein 2; EMS, emergency medical service. ^α^ Australian diabetes society recommended guideline [24]. ^ƛ^ All medications recorded in the case history and discharge referral were recorded for each patient where possible.

**Table 2 ijerph-17-00980-t002:** Summary of notes and requests for ongoing care in discharge referrals. Percentages included only those with discharge referral in the medical record for the admission (n = 837, 95.2%).

Parameter	Frequency (%)
Any diabetes-related note	171 (20.4)
Changes to diabetic medications during admission noted	109 (13.0)
Abnormal blood glucose result noted	104 (12.4)
Request for GP and/or patient to monitor blood sugar level	68 (8.1)
Request for GP to refer patient for HbA1c test	6 (0.7)
Request for GP to follow-up HbA1c result	1 (0.1)
Request for GP to review patient’s diabetes medications	38 (4.5)
Consultation by diabetes educator during admission noted	21 (2.5)
Current HbA1c test result noted	20 (2.4)
Previous HbA1c test result noted	8 (1.0)
Changes to management of diabetes during admission noted	7 (0.8)
Ketosis during admission noted	3 (0.4)
Request for patient to follow-up with endocrinologist noted	53 (6.3)

Abbreviation: HbA1c, glycated hemoglobin.

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
