# Peer review of "Diabetes Detection and Communication among Patients Admitted through the Emergency Department of a Public Hospital"

_ijerph, 2020, doi:10.3390/ijerph17030980_

Round 1

Reviewer 1 Report

-minor - Line 47: institutionalization can be interpreted as psychiatric hospitalization. Suggest use "hospitalization" here to avoid confusion.

-lines 104-106: Does this mean you include all available >60 or a random subset of these records? Please clarify.

-lines 120-121: how were newly diagnosed diabetes cases determine in the ED? Based on the 7% cutoff you state or is that only for patients with already prevalent diabetes?  

-was a standardized data collection form used for data extraction from the charts?

-Are glucose/HbA1C measurements standard for all ED patients or only those with diabetes or suspected? Given that one of your stated aims at the end of the introduction is to identify the number of newly diagnosed diabetes cases in your cohort, it seems that 3 patients is a low number in your cohort for newly identified diabetes. It seems as though there is likely bias here for negative findings since you are not capturing any patient with a glucose/Hba1C value? or more likely, not many have these measurements performed upon admission to the ER. Did you attempt to address these issues in your data collection? I do agree with your discussion which touches on some of this (1st paragraph and limitations paragraph) but could you give some further discussion on these issues?

-Did you have data on high risk subgroups within your sample set (e.g., as stated, Aboriginals are at higher risk).

-Is it possible to present some data regarding the types of patients that make up the top note requests in table 2? Are the patients having medication adjustment rec'd more likely on multiple therapies or is the abnormal glucose value note always given when there is indeed, a result of elevated glucose/HbA1C found?

Author Response

Response to Reviewer’s comments

Reviewer 1

Comments and Suggestions for Authors

-minor - Line 47: institutionalization can be interpreted as psychiatric hospitalization. Suggest use "hospitalization" here to avoid confusion.

Response: Done.

-lines 104-106: Does this mean you include all available >60 or a random subset of these records? Please clarify.

Response: We included all available records of patients aged 60 years and over after randomization was conducted from a pool of patient records (all ages). This was revised for clarity

“…randomization was used to select a representative sample from a pool of patient records (all ages). Only the medical record of patients aged ≥60 years were reviewed and included in this study.

-lines 120-121: how were newly diagnosed diabetes cases determine in the ED? Based on the 7% cutoff you state or is that only for patients with already prevalent diabetes?

Response: In this hospital, newly diagnosed diabetes cases are determined by HbA1c >6.5%. However, 7% is used to indicate good glycaemic control in the admitted patients with diabetes.

-was a standardized data collection form used for data extraction from the charts?

Response: Using ethics approved data collection form, one researcher (FEW) viewed and systematically retrieved required information from the medical record of the selected admissions.

-Are glucose/HbA1C measurements standard for all ED patients or only those with diabetes or suspected? Given that one of your stated aims at the end of the introduction is to identify the number of newly diagnosed diabetes cases in your cohort, it seems that 3 patients is a low number in your cohort for newly identified diabetes. It seems as though there is likely bias here for negative findings since you are not capturing any patient with a glucose/Hba1C value? or more likely, not many have these measurements performed upon admission to the ER. Did you attempt to address these issues in your data collection? I do agree with your discussion which touches on some of this (1st paragraph and limitations paragraph) but could you give some further discussion on these issues?

Response: HbA1c is a recommended test for people with known diabetes or at high risk of diabetes attending the ED of public hospitals in Australia, however, the increasing high rates of hospitalisations and associated cost suggest that interventions are needed urgently. Majority of the admissions come through the ED and with this region being the hotspot for diabetes in NSW, we thought there would be many more people identified with diabetes over a period of one year. Identifying people with diabetes through HbA1c would enable interventions to be initiated to reduce hospitalisations. We have also expanded our discussion based on new results presented.

-Did you have data on high risk subgroups within your sample set (e.g., as stated, Aboriginals are at higher risk).

Response: There were few Aboriginals in this study group, which was again worrying as many may not have identified during admissions.  Since the participants were older people, they were already a high risk group and that was the focus of the study. However, we have also mentioned the proportion of Aboriginals in this study. The relevant section now reads:

The majority of the admitted patients in this study hospital were Australian-born (n=467, 53.1%, including 23 (2.6%) Aboriginal patients)

-Is it possible to present some data regarding the types of patients that make up the top note requests in table 2? Are the patients having medication adjustment rec'd more likely on multiple therapies or is the abnormal glucose value note always given when there is indeed, a result of elevated glucose/HbA1C found?

Response: Thanks for this suggestion. We have presented the result and the relevant section now reads::

Regarding the types of patients that make up the top note requests in Table 2, further analysis revealed that, younger patients (4.2% for 60-69 years) and those with elevated HbA1c (20.5%) during admission, were more likely to have this result noted in their DS compared to the older patients (2.4% and 0.8% for 70-79yrs and ≥80yrs old, respectively; p=0.03) and those with HbA1c ≤7% (6.0%, P=0.003). Having any diabetes related note in the DS was more likely to occur in patients who are taking multiple diabetes medications (33.7%) compared to those who were taking either two (17.7%) or a single medication (16.8%, p<0.0005) for diabetes management. Patients who were taking SGLT2 inhibitors (41.9% vs 18.7%, p<0.0005), DPP4 inhibitors (30.6 vs 17.7%, p<0.0005), sulfonylureas (32.1 vs 16.9%, p=0.0001) and insulin (44.7% vs 11.0%, p<0.0005) were more likely to have notes related to diabetes in their DS, compared to those who were not taking any of the medications (p<0.0005, for all comparison) during admission. Compared to the patients who had no change in their diabetes medications during admission, all those whose medications were changed, had a diabetes related note in their DS (8.5% vs 100%, p<0.0005), as well as those who were referred to a diabetes educator (100% vs 18.4%, p<0.0005) or specialist (27.1% vs 11.9%, p<0.0005) and those with poor glycaemic control (71.1% vs 36.0%, p<0.0005), during admission.

Reviewer 2 Report

Hello,

Thank you to the authors for the opportunity to review their work. It is important work that clearly required intense resources. I have broken my comments into minor and major for the authors' and editors' consideration.

Major:

I don't understand the study design if the goal was to find missed diabetes. The authors' state on line 199 that the main finding was a low new diagnosis rate; however, I am not sure that they can state that the ED has failed with their current findings. For example, isn't it possible that some patients with an existing diabetes diagnosis in this study period had their original diagnosis in the ED prior to this study period? Did the authors consider extracting records for the at-risk population over many years, then assessing if that population is under tested and/or if the testing works when done? More, is it possible (or not) that those with undiagnosed diabetes don't end up at an ED for possible diagnosis (for example, you don't end up at an ED until you have a major event, and before you have a major event you usually have other symptoms that allowed you to be diagnosed in a less severe setting than the ED)? Would it be possible to look at patients admitted to the ED for diabetes-related complications and see if the people had diabetes, and if not, were they tested for diabetes? I understand the purpose of the study was to talk about potential interventions at the ED, but the authors haven't clearly demonstrated there is potential for a large number of new cases with their data (they do it through other studies but not with their analysis). Finally, making a claim about the potential impacts on public policy from 3 patients is likely not robust and re-inforces that maybe testing at the ED won't make a big difference. Admittedly I am not a clinician, but to a biostatistician the result on line 215 seems to be your most defensible and impactful result. This seems like an important result to share, and thus I would ask the authors to confirm they don't want to stress this finding more in the work.  The current manuscript is very dense and reads like the authors wanted to report every possible detail without focusing enough on the structure and how to emphasize key findings (for example, all results seem to get equal weight here and the main study questions get lost). Also, consider visualizations besides just a table (and the supplemental pie chart).

Minor:

I think section 2.3 could be improved. They flowchart is quite nice, but the text description is not as clear. For example, line 102 doesn't say that it is the diabetes only cohort. Also, the fact that exactly '1500' were randomly chosen should be in this section and also justified (For example, did you do a power analysis or was this because of resource limitations (feasibility as currently stated)?). Was there a measure of chart review reliability done? For example, could a second person review 25 charts to get a reliability measure? The statistical analysis section should be improved. For example, please confirm that all assumptions for parametric tests were met. Also, for the GLM, does this mean that all differences in means/proportions referenced in the results were adjusted for age, gender, and ethnicity? Last, did you actually give any aORs in the results? In the results section, how do you derive the 8.9%. This is not 4796/55968. In general, I found the results section to be quite dense. Is it necessary to have all this information in the text (for example, the presentation times)? Can the main results be stressed or emphasized better? Table 1 was similarly hard for me to follow. One idea is to indent subsections. More, the formatting is not consistent throughout. For example, the country of birth provides that results will be (n,%) while the HbA1c provides n (%). Please check the Weight row as well.  Do the authors have a sense about the 879-837 patients without discharge summaries? Consider adding one comment about this. In general, the authors used p-values but didn't provided any confidence intervals for means/proportions. I would advise adding confidence intervals to key results if you are comfortable reporting statistical results related to means/proportions. There are numerous errors in the writing. A partial list includes: Line 47: an 'and' is needed after 'status,' Line 62: clarify 'individuals risk factors for' Line 64: change to a ';' after intervention Line 71: clarify 'closely followed monitored' Line 80: small 'h' in Hospital Line 93: 'records' Line 131: pluralize all statistics Line 135: unique 'people' admitted Figure 1: add arrows to all lines Line 151: missing a '.' before 'Many' Line 167: missing the percent for >=80 years and missing a '(' in front of 'p=0.002'. Line 194: what comparison is this p-value related to? Check all p's are either italicized or not consistently.  Line 201: remind the reader of the age group (don't just say 'this') Line 232: 'confirms'. One note: confirms is a very strong word for an association/quantification study  (same as in line 239) Line 243: fix formatting on '32.2%' Line 245: period needed after 'al' Line 267: did you intend 'suggesting'? Line 268: clarify 'affect management plan of patients'

Author Response

Reviewer 2

Comments and Suggestions for Authors

Hello,

Thank you to the authors for the opportunity to review their work. It is important work that clearly required intense resources. I have broken my comments into minor and major for the authors' and editors' consideration.

Response: We are most grateful to the reviewer for the very useful comments. This has highlighted some of new findings which we have also included in the revised manuscript.

Major:

I don't understand the study design if the goal was to find missed diabetes. The authors' state on line 199 that the main finding was a low new diagnosis rate; however, I am not sure that they can state that the ED has failed with their current findings. For example, isn't it possible that some patients with an existing diabetes diagnosis in this study period had their original diagnosis in the ED prior to this study period? Did the authors consider extracting records for the at-risk population over many years, then assessing if that population is under tested and/or if the testing works when done? 

Response: As a care process measure, the Australian government recommends HbA1c testing every 3 months in people with known diabetes attending the hospitals. In this study, we reviewed all partient records and went back to three months prior to their current admission or visit to the ED if HbA1c notes was not found in recent record. This was also to capture people with earlier diagnosis. The study design was for a period of one year and it was not feasible to extract data going back to years ago because of the hospital is one of the largest hospitals in the Metropolitan Sydney. Someone would also need to manually go through all patients records in the electronic medical record and power charts of patients including any referral notes. This is beyond the scope of this non-funded study.

In addition, the code used for data extraction used by the ICD department, is designed to identify any recorded diagnosis of diabetes and with regards to high risk group, this was the right population due to their older age. pital via the ED within the period of this study. See

A section in the introduction (Lines 53-59) explains why the older people were a high risk group and we have also revised the abstract to make the aims clearer: One example where this was done in the abstract now reads:

“The proportion of admitted patients who were diagnosed with diabetes over a year period, proportion with glycated hemoglobin A1c (HbA1c) and random blood glucose (RBG) test”

The study did not set out to find missed diagnosis, rather as stated in lines 71-76,

More, is it possible (or not) that those with undiagnosed diabetes don't end up at an ED for possible diagnosis (for example, you don't end up at an ED until you have a major event, and before you have a major event you usually have other symptoms that allowed you to be diagnosed in a less severe setting than the ED)?

Response: Yes, this is possible and as noted previously, 90% of the admitted patients in this hospital are triaged through the ED, so we would have captured majority of the cases. There would be few persons who cannot be captured in this data as would any other investigation.

However, we have added a sentence in the limitations that

‘and data for some patients with diabetes who attended the ED and were discharged on the same day may not have been captured in this study

Would it be possible to look at patients admitted to the ED for diabetes-related complications and see if the people had diabetes, and if not, were they tested for diabetes?

Response: Done.

Data for the patients admitted to the ED for diabetes-related complications were further analysed and their uptake of HbA1c during the admission presented in Figure 4.

Data extracted from the ICD department included only people who were coded with diabetes and were admitted through the ED. This should include people who had diabetes related complications during the admission.

I understand the purpose of the study was to talk about potential interventions at the ED, but the authors haven't clearly demonstrated there is potential for a large number of new cases with their data (they do it through other studies but not with their analysis).

Response: In addition to the other additional analysis to demonstrate the need for a potential intervention that could begin by testing people from the ED prior to admission, we have revised the first paragraph of the discussion.

Lines 253-271 now reads: The main findings of this study were the low diagnosis rate of diabetes (0.3%) in the admitted patients coded with diabetes over a one year period, the low communication rate between the hospitals and the general providers, and the very low uptake of glucose monitoring using HbA1c test, even in those with diabetes-related complications/comorbidities, despite the high prevalence of diabetes in this age group[8] and in this region [25]. Since a larger than average number of undiagnosed individuals with diabetes were expected to be living in this area, it is possible that a significant proportion of the individuals with diabetes may have remained undiagnosed throughout their hospital stay due to the low uptake of HbA1c testing during admission and thus, are prone to having more complications due to the hyperglycaemic duration [12, 26]. Despite the various government initiatives encouraging routine screening and monitoring of diabetes in the Australian public hospitals, particularly at the ED, using HbA1c test [27, 28], and the evidence favouring screening for diabetes using HbA1c over other screening techniques including Oral Glucose Tolerance testing (OGTT) [27], this SWS public hospital appear to not be delivering this much needed service, even in such high risk group. The potential medico-legal liability of diagnosing without ensuring follow-up occurs, crowding and staff constraints of the ED and wards, and reluctance of doctors to treat hyperglycemia for fear of causing hypoglycemia are possible barriers to effectively diagnose and prevent diabetes related complications [29, 30]. Initiating an automatic request procedure for HbA1c testing particularly in high risk groups (older adults and Aboriginals) admitted through the ED is needed to reduce the burden of diabetes across the district.

Finally, making a claim about the potential impacts on public policy from 3 patients is likely not robust and re-inforces that maybe testing at the ED won't make a big difference.

Response: We think that in the light of the recent analysis, we have shown that change is needed and adherence to recommended testing procedures will result in more cases identified and properly managed even during admissions. No change was made

Admittedly I am not a clinician, but to a biostatistician the result on line 215 seems to be your most defensible and impactful result. This seems like an important result to share, and thus I would ask the authors to confirm they don't want to stress this finding more in the work. 

Response: Thanks for the suggestion. Following the first reviewer’s comments, we have added new results from analysis of DS which also highlights this section

“The fact that the important HbA1c test result or request related to the patient’s diabetes was made only in one of five admitted patient records indicate the need to measure and improve discharge summary quality in this hospital. Information in the discharge summary helps the general practitioner in decision making including altering a patient’s management plan, and/or referring to specialist (endocrinologist, dietician or diabetes educator). In this study, participants aged between 60 - 69 years who have inadequate glycemic control and were referred to a diabetes specialist were more likely to have these recorded in their discharge summary notes. For example, in all three participants who were diagnosed with diabetes during their admission, this was noted in their discharge summaries, along with newly prescribed medications. Ideally, once a patient is diagnosed with diabetes or found to have blood glucose or HbA1c results suggestive of diabetes, this should be noted with a request for follow-up in the discharge referral and absence of such records may compromise the ability for treatment and prevention of complications [17, 29, 30], which may result in a higher risk of re‐admission [31].”

The current manuscript is very dense and reads like the authors wanted to report every possible detail without focusing enough on the structure and how to emphasize key findings (for example, all results seem to get equal weight here and the main study questions get lost). Also, consider visualizations besides just a table (and the supplemental pie chart).

Response: Thanks for the suggestion. The result has been reduced in text and more visualizations were used. There are now four figures in the manuscript.

Minor:

I think section 2.3 could be improved. They flowchart is quite nice, but the text description is not as clear. For example, line 102 doesn't say that it is the diabetes only cohort.

Response: The text description has been made clearer. The referred line now reads:

The extraction included 5,100 electronic admission records of patients who visited the ED within the study period and were coded with diabetes

Also, the fact that exactly '1500' were randomly chosen should be in this section and also justified (For example, did you do a power analysis or was this because of resource limitations (feasibility as currently stated)?).

Response:  No power analysis was done since data for all patients, who were admitted through the ED was obtained. It was feasible to select 1500 patients from the random pool of 4796 as we needed to manually review all patient medical records to retrieve other needed information.

Was there a measure of chart review reliability done? For example, could a second person review 25 charts to get a reliability measure?

Yes, there was. A second person, who reviewed the medical record of 35 patients. Since there was a template for use, there was no variation in the data extracted by both team members.

The statistical analysis section should be improved. For example, please confirm that all assumptions for parametric tests were met.

Response: Done.

Normality distribution assessment confirmed that all assumptions met the parametric text.

Also, for the GLM, does this mean that all differences in means/proportions referenced in the results were adjusted for age, gender, and ethnicity?

Response: Yes. But no difference was found between variables.

Last, did you actually give any aORs in the results?

Response: We have now given the adjusted odds ratio where significant associations were found in the multivariable analysis. Other variables were not associated with Hba1c above 7%.

In the results section, how do you derive the 8.9%. This is not 4796/55968.

Response: This was a typo error and has been corrected to read 8.6%.

In general, I found the results section to be quite dense.

Is it necessary to have all this information in the text (for example, the presentation times)?

Response: The presentation times have been removed to make the result section easier to comprehend and other sections in the result were also revised for clarity and to enhance the flow such as: Of complications, nephropathy was recorded in 29.5% of the participants’ record and at least one comorbidity was present in 87% of the admitted participants with diabetes.

Can the main results be stressed or emphasized better?

Response: The results have been made clearer by emphasizing the relevant sections in addition to deleting some of the information that were not important such as the presentation times. In addition, we have provided sub-headings to highlight the various results and included an additional analysis of the discharge summary notes as requested by the first reviewer.

Table 1 was similarly hard for me to follow. One idea is to indent subsections. More, the formatting is not consistent throughout. For example, the country of birth provides that results will be (n,%) while the HbA1c provides n (%). Please check the Weight row as well. 

Response: All recommended formatting was done. Table 1 was also revised for clarity. Some variables included in text

Do the authors have a sense about the 879-837 patients without discharge summaries? Consider adding one comment about this.

Response: As shown in Table 2, 837 patients had a discharge summary, however, only few had any diabetes related notes in their DS (n=171).

In general, the authors used p-values but didn't provided any confidence intervals for means/proportions. I would advise adding confidence intervals to key results if you are comfortable reporting statistical results related to means/proportions.

Response: We have provided confidence intervals where relevant in this study. See section on Hba1c above 7%.

There are numerous errors in the writing. A partial list includes:

Line 47: an 'and' is needed after 'status,'

Line 62: clarify 'individuals risk factors for' Line 64: change to a ';' after intervention

Response: Revised ‘There are many individuals with risk factors for diabetes..’

“The ED also provides medical care to people with little or no access to screening and preventive interventions, therefore, initiating a screening interventions in the ED may improve the health of this vulnerable population”

Line 71: clarify 'closely followed monitored'

Response: 'closely monitored' has been used

Line 80: small 'h' in Hospital Line 93: 'records'

Response: Done

Line 131: pluralize all statistics Line 135: unique 'people' admitted

Response: done

Figure 1: add arrows to all lines Line 151: missing a '.' before 'Many'  

Response: Done

Line 167: missing the percent for >=80 years and missing a '(' in front of 'p=0.002'.

Response: Done

“among the admitted participants who were aged 60-69yrs (4.3%) compared to those who were aged 70-79yrs (1.3%) and those aged ≥80 years and over (0.4%, p=0.002).

Line 194: what comparison is this p-value related to? Check all p's are either italicized or not consistently.  Line 201: remind the reader of the age group (don't just say 'this')

Response: Done

“For the participants with above threshold HbA1c, 20% had a note regarding the test result in their discharge summaries, compared to 5.8% (p=0.015) of the participants with HbA1c ≤7% (n=3) who had such notes in their discharge summaries”

Line 232: 'confirms'. One note: confirms is a very strong word for an association/quantification study (same as in line 239)

Response: Suggests has been used in both instances.

Line 243: fix formatting on '32.2%' Line 245: period needed after 'al'

Response: done

Line 267: did you intend 'suggesting'?

Response: done

Line 268: clarify 'affect management plan of patients' 

Response: Unavailability of such notes in the discharge summary may affect the management plan of patients or lead to unnecessary test repeats during consultation

Round 2

Reviewer 1 Report

The authors have done a good job of responding to reviewer comments and making revisions accordingly. I have no further comments

Author Response

Reviewer 1

The authors have done a good job of responding to reviewer comments and making revisions accordingly. I have no further comments

Response: We thank the reviewer for the very useful comments

Reviewer 2 Report

Thank you to the authors for their detailed responses. The revised manuscript is much more readable and the authors' goals are much clearer. I have a few minor points remaining, but I understand if the editors override my concern.

There are still typos in the abstract and in some of the changed material. For example, capitalization on line 37 and no period on line 41. Also, is it supposed to be 8.6% and not 8.9% on line 23?  What prevalence (%) did you expect if not 8.6% (what is the expected prevalence)? Why didn't you do a statistical test for a difference between 8.6% and the expected percent (or at least report the difference if you are thinking of your data as a full census of the ED visits in the year)? This is the major point of the manuscript and I don't think it is in the text. I realize now (unless I am wrong) that the main argument is that with the 3 month chart review (consider adding this detail to the manuscript), all individuals with known diabetes should be identifiable by the authors with the present design. Thus, you based everything on this assumption, and measured if the percent with diabetes in the ED was significantly lower than the expected percent in the general population (and if so, then there must be people going to the ED with unknown diabetes that aren't being identified with current practices). The study design didn't really let you find potentially missed diabetics as you conditioned your numbers on those coded diabetic in the ED. In my opinion, it is a big leap to say that the lower percent of diabetics in the ED than expected means there are people going to the ED that are diabetic but not identified. For purposes of this review, I think it is difficult to support publishing a study where this main conclusion hasn't really been tested directly. For example, with only 3 new cases, couldn't an argument be made that there really aren't many new cases to be found (e.g. someone could say that even with very inconsistent testing, they would have found more than 3 out of the other 91.4% of ED visits if there really was more diabetes)? In fairness, the authors present their findings only as evidence that potentially shows a way to improve diabetes identification; however, I think the discussion should, at a minimum, more clearly state the limitations related to the study design.  Note: the authors cite statistical methods (e.g. ANOVA) that require more than the normality assumption. Also, discussing the reliability of the review could be more exact (line 126: 'similar'). Please define 'affected' in Figure 4 Line 249: How can there be an at-risk population if no demographics are related to elevated HbA1C? Not in any way saying this is wrong, but it surprised me to read this (maybe because you already restricted to the at-risk population at least by age?). I got a little lost in understanding the new medication based results/interpretations. However, it seems like these were related to the other reviewer's comments; so, I will defer to that reviewer.

Author Response

Reviewer 2

Thank you to the authors for their detailed responses. The revised manuscript is much more readable and the authors' goals are much clearer. I have a few minor points remaining, but I understand if the editors override my concern.

There are still typos in the abstract and in some of the changed material. For example, capitalization on line 37 and no period on line 41.

Response: line 37 -41 was not part of the revised version and was deleted.

To improve comprehension, we have also made other revisions in language across the manuscript, which we highlighted in red fonts.

Also, is it supposed to be 8.6% and not 8.9% on line 23? 

Response: The value was supposed to 8.6% and has been corrected

What prevalence (%) did you expect if not 8.6% (what is the expected prevalence)? Why didn't you do a statistical test for a difference between 8.6% and the expected percent (or at least report the difference if you are thinking of your data as a full census of the ED visits in the year)? This is the major point of the manuscript and I don't think it is in the text.

Response: We expected a higher prevalence of diabetes in this region based on the recent monograph showing a 15% population prevalence aged 60 years and over and the report from a neighbouring district with similar demography and prevalence of diabetes, where the authors found 32.2% prevalence in adults that were tested for Diabetes at the ED using HbA1c,over 6 months. We have included these arguments in the recent revision.

“A higher prevalence of diabetes (38.4% vs 8.6%) was found in another public hospital ED in Western Sydney, a neighboring district with similar high diabetes prevalence, and 32.2% of those with diabetes were previously undiagnosed diabetes of diabetes (32.2%), after 6 months of testing for diabetes in adults using HbA1c, irrespective of their presenting signs[12]. The proportion of newly diagnosed individuals with diabetes increased by 7 folds (from 0.3% - 2.2%) considering only the individuals with HbA1c test done during the admission (n=3/137), however, this was still lower than the previous study[12] and the 15% population prevalence of diabetes reported in the recent monograph in individuals aged ≥60years [8].”

I realize now (unless I am wrong) that the main argument is that with the 3 month chart review (consider adding this detail to the manuscript), all individuals with known diabetes should be identifiable by the authors with the present design. Thus, you based everything on this assumption, and measured if the percent with diabetes in the ED was significantly lower than the expected percent in the general population (and if so, then there must be people going to the ED with unknown diabetes that aren't being identified with current practices). The study design didn't really let you find potentially missed diabetics as you conditioned your numbers on those coded diabetic in the ED.

Response: We have considered your suggestion in the latest revision. Sentences added to the abstract, discussion and limitation sections to reflect this suggestion include:

Abstract: ….and/or are going to the ED with unknown diabetes that are unidentified with current practices.

First paragraph in discussion: “The percentage of individuals with diabetes in this study was significantly lower than the expected proportion in the general population, which indicates that there must be people either going to the ED with unknown diabetes that are unidentified with current practices, and/or..

And in the limitation section : The 3-month chart review should have identified all individuals with known diabetes, but in the absence of testing, the study design restricted the data to participants coded with diabetes in the ED, making it difficult to find potentially missed cases with diabetes.

In my opinion, it is a big leap to say that the lower percent of diabetics in the ED than expected means there are people going to the ED that are diabetic but not identified. For purposes of this review, I think it is difficult to support publishing a study where this main conclusion hasn't really been tested directly. For example, with only 3 new cases, couldn't an argument be made that there really aren't many new cases to be found (e.g. someone could say that even with very inconsistent testing, they would have found more than 3 out of the other 91.4% of ED visits if there really was more diabetes)?

Response: This is not a big leap in the light of recent evidence from Western Sydney public hospital (an area with similar demography and rate of diabetes), which showed 38.4% prevalence of diabetes after 6 months of opportunistic testing using HbA1c at the ED (32.2% of those with diabetes were previously undiagnosed). In addition, the recent SWS diabetes monograph reported a 15% population prevalence of diabetes in those aged 60years and over. The entire section in the first paragraph of the introduction was revised and now reads like so:

“A higher prevalence of diabetes (38.4% vs 8.6%) was found in another public hospital ED in Western Sydney, a neighboring district with similar high diabetes prevalence, and 32.2% of those with diabetes were previously undiagnosed diabetes of diabetes (32.2%), after 6 months of testing for diabetes in adults using HbA1c, irrespective of their presenting signs[12]. The proportion of newly diagnosed individuals with diabetes increased by 7 folds (from 0.3% - 2.2%) considering only the individuals with HbA1c test done during the admission (n=3/137), however, this was still lower than the previous study[12] and the 15% population prevalence of diabetes in those aged ≥60years [8]. Since this region and Western Sydney are the hotspots for diabetes in the country[26], a larger than average number of undiagnosed individuals with diabetes were expected to be living in these areas. The percentage of individuals with diabetes in this study was significantly lower than the expected proportion in the general population, which indicates that there must be people either going to the ED with unknown diabetes that are unidentified with current practices, and/or that a significant proportion of the individuals with diabetes remain undiagnosed throughout their admission. This may be due to the low uptake of HbA1c testing during admission and thus, are prone to having more complications due to the hyperglycaemic duration [13, 27]. It appears that this SWS public hospital may not be delivering this much-needed service, even in such high-risk age group[8]

In fairness, the authors present their findings only as evidence that potentially shows a way to improve diabetes identification; however, I think the discussion should, at a minimum, more clearly state the limitations related to the study design. 

Response: This has been included in the limitation. See comment above

Note: the authors cite statistical methods (e.g. ANOVA) that require more than the normality assumption.

Response: One-way analysis of variance was performed in SPSS only after all test conditions reuired for such testing including normality test were satisfied.

Also, discussing the reliability of the review could be more exact (line 126: 'similar').

Response: This was done and the section now reads: “…and the data extracted was the same as that extracted by the first reviewer”

Please define 'affected' in Figure 4 Line 249:

Response: Done. We have also replaced the two abbreviated terms (HTN and IHD) with the full definitions.

How can there be an at-risk population if no demographics are related to elevated HbA1C? Not in any way saying this is wrong, but it surprised me to read this (maybe because you already restricted to the at-risk population at least by age?).

Response: The age group of the participants is considered at-risk population for diabetes and this has been clarified in the discussion, “A formal early detection of diabetes through opportunistic testing in this high-risk age group population”

I got a little lost in understanding the new medication based results/interpretations. However, it seems like these were related to the other reviewer's comments; so, I will defer to that reviewer. 

Response: The results were in relation to the other reviewer’s comments.
